# Functional Proteomics of Nuclear Proteins in *Tetrahymena thermophila*: A Review

**DOI:** 10.3390/genes10050333

**Published:** 2019-05-01

**Authors:** Alejandro Saettone, Syed Nabeel-Shah, Jyoti Garg, Jean-Philippe Lambert, Ronald E. Pearlman, Jeffrey Fillingham

**Affiliations:** 1Department of Chemistry and Biology, Ryerson University, 350 Victoria Street, Toronto, ON M5B 2K3, Canada; alejandro.saettone@ryerson.ca; 2Department of Molecular Genetics, University of Toronto, 1 King’s College Circle, Toronto, ON M5S 1A8, Canada; nabeel.haidershah@mail.utoronto.ca; 3Department of Biology, York University, 4700 Keele Street, Toronto, ON M3J 1P3, Canada; jyoti@yorku.ca (J.G.); ronp@yorku.ca (R.E.P.); 4Department of Molecular Medicine and Cancer Research Centre, Université Laval, Quebec, QC G1V 0A6, Canada; Jean-Philippe.Lambert@crchudequebec.ulaval.ca; 5CHU de Québec Research Center, CHUL, 2705 Boulevard Laurier, Quebec, QC G1V 4G2, Canada

**Keywords:** chromatin biology, nuclear processes, proteomics, affinity purification, mass spectrometry, protists, ciliates

## Abstract

Identification and characterization of protein complexes and interactomes has been essential to the understanding of fundamental nuclear processes including transcription, replication, recombination, and maintenance of genome stability. Despite significant progress in elucidation of nuclear proteomes and interactomes of organisms such as yeast and mammalian systems, progress in other models has lagged. Protists, including the alveolate ciliate protozoa with *Tetrahymena thermophila* as one of the most studied members of this group, have a unique nuclear biology, and nuclear dimorphism, with structurally and functionally distinct nuclei in a common cytoplasm. These features have been important in providing important insights about numerous fundamental nuclear processes. Here, we review the proteomic approaches that were historically used as well as those currently employed to take advantage of the unique biology of the ciliates, focusing on *Tetrahymena*, to address important questions and better understand nuclear processes including chromatin biology of eukaryotes.

## 1. Expression, Functional and Comparative Proteomics

The term proteome describes the complete set of proteins in a cell, organelle or organism [1]. Proteomics refers to the study of proteins encompassing the identification, quantification, and characterization of their composition, concentration, structure and activity. Several sub-fields have emerged including expression, functional, and comparative proteomics [2,3]. Expression proteomics refers to the identification and differential analysis under different stimuli/developmental stages/disease stage of all proteins present in a particular cell/organelle/organism [2]. More recently, these types of proteomic analyses have been extended to include the cataloguing of post-translational modifications (PTMs) on proteins. The importance of assemblies of proteins into functional units (protein complexes) to the cell in directing cellular function is well established [4]. Functional proteomics as an experimental approach relies on “guilt by association” [2]. If a protein of unknown function is shown to physically interact with a protein of known function, it can be inferred as a first approximation that the proteins have similar function and specific hypotheses can be derived about the function of the first protein. Affinity purification (AP) exploits the high specificity of antibody-antigen interactions or other affinity reagents by allowing selective purification via an antibody-conjugated chromatographic medium or other affinity matrix of its cognate epitope tag that has been attached to the ‘bait’ protein. Co-purifying proteins that physically interact with the ‘bait’ protein, the so-called set of ‘prey’ proteins, may then be identified via various means, most commonly mass spectrometry (MS). Affinity purification and mass spectrometry (AP-MS) when combined with high throughput methods for the systematic discovery of protein-protein interaction (PPI), and combined with bioinformatic analysis [5,6,7,8] has led to the development of comprehensive interaction maps for proteins from many organisms including the prokaryote *Escherichia coli* [9] in addition to eukaryotes such as the yeast *Saccharomyces cerevisiae* [10], and human cells [11]. Comparative studies of protein–protein interaction networks among the limited number of eukaryotic species for which proteomic data exists show that many circuits embedded within the protein networks are conserved over evolution [12]. Comparative proteomics has the potential to permit functional annotation of evolutionarily conserved proteins of unknown function [13]. 

## 2. *Tetrahymena thermophila*: A Useful Proteomic Model for Nuclear Events

Most model species exploited for proteomic analysis to date belong to either the prokaryotes [14], fungi (particularly yeast) [10], and animals such as *Drosophila* [15], worm [16], and human [11], members of the Opisthokont lineage of eukaryotes. Large-scale PPI data gaps currently exist for a number of other eukaryotic lineages including protists, and green plants. The protist *T. thermophila* is a ciliate, a member of the Alveolate group of eukaryotes, along with dinoflagellates and the parasitic apicomplexans that include *Plasmodium* species that cause malaria. *Tetrahymena* is a unicellular eukaryote with a variety of unique biological features that have made it a powerful model system for many aspects of molecular and cellular biology, particularly with respect to nuclear biology where it segregates germ-line specific transcriptionally silent, and somatic transcriptionally active chromatin into two structurally and functionally distinct nuclei contained within its single cell [17]. The micronucleus (MIC) is diploid, divides by mitosis and is not transcribed during growth. In addition, the MIC undergoes meiosis during the sexual phase of the life cycle, conjugation, and is analogous to a germ-line nucleus. The macronucleus (MAC) is polyploid, divides amitotically, is transcriptionally active and is not inherited sexually, an analog of a somatic nucleus. The MAC is rich in acetylated chromatin which was critical in the discovery of the first lysine acetyl transferase (KAT) p55/Gcn5 [18,19]. During the conjugation (Figure 1), genome-wide transcription of non-coding RNAs (ncRNAs) occurs that directs large-scale RNAi-dependent assembly of distinct chromatin domains in the new MAC as a prelude to programmed DNA rearrangements including site-specific DNA deletion, analogous to irreversible gene silencing [20] that are similar to the enigmatic developmentally regulated chromatin diminution that occurs in presomatic cells of the parasitic nematode Ascaris [21]. *Tetrahymena* grows axenically with a generation time of approximately 2.5 h in inexpensive rich media [22] and large-scale culture is routine. The MAC and MIC genomes are sequenced [23,24] and molecular genetic methods developed for *Tetrahymena* include homologous recombination mediated exact gene replacement in the MAC or MIC allowing gene knockout and epitope tagging [25], in addition to inducible RNAi gene knock down [22,26,27]. The Tetrahymena Genome Database (TGD and its associated Wiki, www.ciliate.org), contains the MAC and MIC genome sequences as well as predicted gene models [28,29]. The Tetrahymena Functional Genomics Database (TetraFGD, http://tfgd.ihb.ac.cn) contains functional genomic data sets relating to gene expression in addition to gene networks [30,31]. This review focuses on the past, present and future utilization of proteomic approaches in the *Tetrahymena* model to advance knowledge of diverse nuclear events such as chromosomal replication, genome rearrangement, genome stability, transcription, and chromatin remodeling. 

## 3. Identification of Core Histones and their Variants

The biochemical identification and characterization of *Tetrahymena* core and variant histones are early examples of the utility of expression proteomics. After developing a differential centrifugation-based method to separate MACs and MICs [32], Gorovsky compared acid-soluble fractions enriched in positively charged DNA-binding chromatin proteins from the two nuclei [33]. Using one dimensional (1D) SDS-PAGE, they observed differences between the two nuclei, providing the first indications that their chromatin composition differed. Allis et al. [34], extended the study by using two dimensional (2D) SDS-PAGE to look at differences among the chromatin proteins and found that the MIC contain two types of histone H3 (H3_S_ and H3_F_), one with a slightly faster (H3_F_) gel mobility that was subsequently found to be a consequence of a regulated proteolytic event that removes the first six amino acids from the H3 N-terminus [35]. Other acid-soluble MAC proteins identified in this type of analysis include histone variants hv1 (H2A.Z) and hv2 (H3.3) [36], a finding that at the time provided important information concerning the function of these conserved eukaryotic proteins in transcription. When MAC and MIC acid-soluble nuclear proteins were further separated into perchloric acid (PCA) soluble and insoluble fractions, 1D SDS-PAGE expression proteomic analysis let to the discovery of two high mobility group (HMG) proteins [37], distinct MAC and MIC specific histone H1 proteins [37,38], and a MAC-specific HP1 [39]. The MIC specific H1 is unique in that it is processed into different peptides (α, β, γ, and δ) [38]. To date the function of regulated processing events (MIC-specific H1 and H3_F_), as well as the identity of the respective processing protease(s), is unknown. 

## 4. Telomeres and Telomerase

Proteomic analysis in *Tetrahymena* has yielded information concerning the nature of telomerase. Telomeres are repetitive DNA sequences located at the ends of linear chromosomes, which are required for stability and replication [40]. It was reasoned that the large number of telomeres in the *Tetrahymena* MAC should require an amount of telomerase that could be detected in cell extracts. Using a novel primer extension based assay, a telomere terminal transferase activity was identified in nuclear extracts from conjugating cells [41]. Gel-filtration and ion exchange chromatography were used to partially purify the activity and demonstrate the existence of the co-purifying RNA component [42]. A similar approach led to the identification of regulatory proteins p80 and p95 that co-purified with the activity [43]. The catalytic subunit proved recalcitrant to identification until similar analysis was performed using a hypotrichous ciliate with millions of gene-sized MAC DNA molecules. The then newly developed nanoelectrospray tandem MS was used to analyze the fraction containing telomerase activity resulting in the identification of p123, the catalytic subunit of telomerase that contained reverse transcriptase motifs [44] whose orthologs were soon identified using comparative sequence analysis to design degenerate primers which led to the molecular cloning of the cDNA and genomic locus of TERT in *Tetrahymena* [45]. With TERT identified, a rapid and gentle AP strategy based on the newly developed tandem affinity purification (TAP) method of Rigaut et al. [46] was utilized [47] to affinity purify the complex in order to assess the full protein complement. TAP-tagged TERT was integrated into the *Tetrahymena* genome into the non-essential β-tubulin 1 (BTU1) locus, replacing the coding region in the strain Cu522, a strain engineered so that complete replacement of all copies of BTU1 results in resistance to taxol. Genes inserted into the locus are over-expressed under the control of the strong *BTU*1 promoter [48]. To demonstrate functionality of the C-terminal tagged TERT, the gene encoding WT *TERT*, which was previously demonstrated to be essential, was deleted from TERT-TAP (*BTU*1) and shown that TERT-TAP could completely replace function of the essential WT TERT [47]. SDS-PAGE analysis of the tandem affinity purified TERT-TAP revealed four major polypeptides (p20, p45, p65 and p75) in addition to p123/TERT. Molecular analysis of p45 and p65 demonstrated essential roles in the maintenance of telomere length [47,49] showing the validity of the approach. Initial attempts to obtain peptide sequences from gel slices containing p75 and p20 were unsuccessful, therefore the authors repeated the AP as a one-step purification to increase yield for sequencing by Edman degradation and were able to identify the corresponding cDNAs [50]. Reciprocal purification of over-expressed ZZ-p75 (ZZ epitope tag in tandem protein A domains followed by TEV cleavage site) in a Cu522 background identified p123, p65, p45 and p20, and its genetic analysis revealed a role in telomere homeostasis [50]. For these and subsequent AP, MS was used to identify co-purifying proteins guided by gene predictions from the *Tetrahymena* macronuclear genome sequencing project [23] and annotated in the *Tetrahymena* Genome Database (TGD; www.ciliate.org) [28]. Subsequent AP identified a ubiquitin ligase as the major interacting partner of p20, consistent with its molecular identity as the Skp1 component of SCF (Skp1/Cullin/F-box protein) ubiquitin ligase suggesting that the telomerase-p20 interaction was a result of over-expression of telomerase subunits [51]. A C-terminal FZZ (3xFLAG—TEV cleavage—FZZ) epitope tag was added to p123/TERT at the endogenous locus so that the tagged protein’s expression was dependent on the gene’s endogenous promoter. The set of p123 co-purifying proteins included p45, p65 and p75 identified previously, in addition to three new proteins, p19, p50 and p82/Teb1 that is paralogous to the Rpa1 large subunit of the general single-stranded DNA binding heterotrimer replication protein A (RPA) [51]. Finally, the existence of two additional proteins in a functional telomerase complex was predicted by cryo-EM studies of telomerase before their subsequent identification as Teb2 and Teb3 when TERT-FZZ was affinity purified and analyzed by the more sensitive liquid chromatography–tandem mass spectrometry (LC-MS/MS) [52]. AP of Teb2 and Teb3 both co-purified low levels of telomerase activity and subsequent MS identified several telomerase holoenzyme proteins confirming that Teb1, Teb2, and Teb3 forms a module within telomerase [53]. Unlike Teb1, subsequent AP-MS of epitope tagged Teb2 and Teb3 revealed them to not be telomerase-specific, with Rpa1 and each other as their major binding proteins leading to the suggestion that Rpa1, Teb2, and Teb3 together compose *Tetrahymena* RPA [53]. 

The identification of proteins that bind the *Tetrahymena* telomere began with Pot1a, identified in 2007 by its similarity with that of the human orthologue [54]. MS was used to analyze proteins that co-purified with TAP-Pot1, expressed from its endogenous locus. MS analysis of a one-step AP from nuclear extracts found two co-purifying proteins Tpt1 (TPP1/Tpz1 in *Tetrahymena*) and Pat1 (Pot1-associated *Tetrahymena*) [55]. To confirm these interactions, cell lines were engineered expressing either TAP-tagged Tpt1 (TAP-Tpt1) or Pat1 (TAP-Pat1), recombined into the respective endogenous gene locus but expressed under the heterologous cadmium-regulated metallothionein (MTT1) promoter which was done in order to insert the selectable marker upstream of each gene due to N-terminal placement of the tag. Conventional pull-downs were performed using Ni-agarose to purify 6x-HIS tagged proteins from whole cell extracts and an antibody that recognizes Pot1 was used in Western blotting of SDS-PAGE to demonstrate the interaction. Conditional knockouts of either genes affected telomere length demonstrating their importance to telomerase activity [55]. Another study used MS to identify proteins that co-purify with TAP-Tpt1 and TAP-Pat1 [56]. This identified peptides from Tpt1, Pat1, Pot1a, and a new protein named Pat2 (Pot1a-associated *Tetrahymena* 2), suggesting that the proteins present at the 3′ overhangs of *Tetrahymena* telomeres exist in a four-protein complex consisting of Pot1a, Tpt1, Pat1, and Pat2. The detailed mechanism by which this complex regulates telomere length remains to be determined. 

## 5. Programmed DNA Rearrangements

During MAC development (Figure 1), extensive programmed genomic rearrangements occur including fragmentation of chromosomes and concomitant addition of telomeres, DNA amplification, and site specific deletion of ~10–15% of the original genome, comprising ~6000 individual deletion events [57] (Figure 2). Proteomics has generated novel insights into the mechanism of these programmed DNA re-arrangements. Using an expression proteomic approach, Madireddi et al. [58] briefly pulsed conjugating cells with a radio-labeled amino acid before separating enriched fractions of MIC, old MACs, and new developing MAC (Anlagen). Fluorography of proteins separated on 1D SDS-PAGE identified four abundant proteins (p32/Pdd3, p43/Pdd2, p65/Pdd1 and p85) specifically in the Anlagen, but not in the MIC or the parental MAC [58]. Subsequent molecular cloning and functional analysis of the genes encoding Pdd1, Pdd2 and Pdd3 linked them to the heterochromatin formation and programmed DNA deletion during MAC development [59,60,61]. Although not conclusively addressed in the literature, it is probable that the fourth protein, p85, is Twi1, a PIWI/PAZ protein that was found independently to link the RNAi pathway to heterochromatin formation and programmed DNA deletion [62,63]. A limitation of these analyses, which occurred prior to the general use of MS for proteomic analysis, was that a relatively large amount of material was required to sequence peptides of a particular protein in order to be able to design primers and to clone the corresponding gene. We suggest many proteins involved in developmentally programmed DNA rearrangements remain to be identified using more sensitive current MS-based approaches. To address this gap, we have undertaken a comprehensive mass spectrometric analysis of the proteomes of MAC, MIC and development-specific Anlagen (Pearlman et al., unpublished observations). We have modified the 1g ‘stay put’ procedure developed by Allis and Dennison [64] to obtain large quantities of high purity MAC and MIC from vegetative and starved cells, and Anlagen from conjugating cells suitable for MS analysis and used iTRAQ [65] to quantitatively compare protein identification data from the four samples. When ranked by enrichment, top Anlagen proteins included Pdd1, Pdd2, and Twi1 indicating validity of the approach, in addition to several additional proteins expressed in a conjugation-specific manner whose function is being addressed by our functional proteomic pipeline [7]. Ratios of protein amounts in each of the three MAC preps to MIC and the reciprocal identified potential MAC-specific and MIC-specific proteins. These data from complete nuclear proteome analysis will provide insight into questions about functional differences of proteins restricted to one nucleus. This quantitative approach will identify Anlagen-specific proteins and permit a comprehensive understanding of the compositional differences between *Tetrahymena* somatic and germinal nuclei which will allow functional inferences about nuclei-specific proteins.

Twi1 was initially identified and its gene cloned based upon its similarity to members of the PPD gene family defined by conserved PAZ and Piwi domains [62]. Somatic knockout analysis of *TWI1* functionally implicated it in the production of conjugation specific small ~28 nt RNAs named scan RNA (scn RNA) and programmed DNA deletion [62]. The scan RNA model was developed to account for the phenotype of the *TWI1* knockout [62] linking scan RNAs, Twi1 and heterochromatin and excision of IESs, while accounting for their previously demonstrated epigenetic regulation [66]. A functional proteomic approach was adopted to further understand how Twi1p and scnRNA interact [62]. AP-MS of Flag-HA-Twi1p expressed from its endogenous locus during conjugation, identified *EMA1*, a DExH box RNA helicase [67]. Functional analysis of *EMA1* showed it to be required for selective down-regulation of scnRNAs homologous to MAC-destined sequences which are then unavailable to template deletion when transported to the anlagen. Three additional proteins were found to co-purify with Flag-HA-Twi1p. Wag1 and CnjB interact with Twi1 and each other [68]. Interestingly, *CNJB* had previously been identified by sequencing of a conjugation-specific cDNA library [69]. Molecular analysis of *CNJB* and *WAG1* demonstrated that expression of both was required for proper excision of selected IESs [68]. The final Flag-HA-Twi1 interacting protein Giw1 was found by performing AP-MS using gentle cell lysis prior to AP favoring the isolation of cytoplasmic specific proteins [70] and its molecular analysis demonstrated a function in Twi1 nuclear localization [70]. AP-MS of Pdd1, Pdd2 or Pdd3 has not been reported but will likely result in the identification of new proteins involved in IES deletion/programmed DNA rearrangements/irreversible genome silencing. 

## 6. RNAi Mechanisms

Proteomic approaches have led to new insights into fundamental RNAi mechanisms. After identifying a class of small RNAs (sRNAs) of 23 and 24 nt that are constitutively expressed [71] as opposed to the developmental specific class of 27–30 nt scnRNAs, an AP-MS strategy was adopted by the Collins lab in order to obtain information about sRNA biogenesis. Previous work identified two orthologs of Dicer (Dcl1 and Dcl2), as well as one RNA dependent RNA polymerase (Rdr1). Unlike a C-terminal version engineered at its homologous locus, an N-terminal version over-expressed from *BTU1* in Cu522 was found to replace the essential function of Rdr1 [72]. AP-MS identified a set of four co-purifying proteins (Rdn1 and Rdn2 migrating ~65kD and Rdf1 and Rdf2 migrating ~40 kD on a silver stained gel) that had similarities and differences from proteins of the RNA-dependent RNA polymerase complex (RDRC) in *Schizosaccharomyces pombe*, a powerful model for RNAi studies [72]. AP-MS when performed using low stringency washes identified Dcl2 as an Rdr1 interacting protein suggesting RDRC recruits Dcl2. Dcr2 subsequently engineered to express a C-terminal tag from its endogenous locus reciprocally purified RDRC members when performed with low stringency washes [72]. A subsequent study focusing on the role of the other members of RDRC revealed that Rdn2 had a different gene expression profile than Rdr1, Rdn1, Rdf1 and Rdf2 as assessed by Northern blotting [73]. Consistent with the different gene expression profiles, AP-MS of Rdn1-FZZ and Rdn2-FZZ revealed two Rdr-containing complexes, one containing Rdr1, Rdn1, Rdf1 and Rdf2 from starved cells and the other containing Rdr1 and Rdn2 from conjugating cells [73]. Tagged Rdn1 also co-purified a protein of ~120 kDa (Rsp1) with no obvious similar primary sequence motifs or homology with other proteins, was not co-purified by Rdn2, Rdr1 or Dcr2 [73] and was found to function upstream of RDRC and Dcr2 [74].

PAZ/PIWI domain (PPD) proteins carrying sRNAs function in gene and genome regulation. *Tetrahymena* encodes twelve PPD domain-containing proteins, eight of which were overexpressed in the *BTU*1 locus in strain Cu522 with N-terminal tandem protein A epitope tags [75] in order to characterize bound sRNAs. Twi12 was the only PPD domain protein whose gene was found to be required for growth [75]. RNA-Seq analysis of RNAs co-purifying with affinity purified FZZ-Twi12 revealed it to associate with RNA fragments derived predominantly from the 3′ end of mature transfer RNAs (tRNAs) [76]. A strain was engineered expressing full-length ZZF-Twi12 from its endogenous locus and was found to localize to the MAC. AP-MS of ZZF-Twi12 revealed two proteins to co-purify including 5′ to 3′ exonuclease Xrn2, both of which when reciprocally analyzed co-purified Twi12 [77]. All three proteins were found by indirect immunofluorescence analysis (IF) to localize to the MAC and function in RNA processing. Further work will be necessary to confirm and expand these observations.

## 7. Epigenetics

Proteomic studies have revealed insight into the identity of histone PTMs. In a 1985 study, conjugating cells were pulse-labeled with [^3^H] lysine for 30 min from 5.0 to 5.5 h of conjugation, and MIC were purified, acid-extracted, and histones analyzed on long acid-urea gels by fluorography [78]. It was observed that newly synthesized histone H3 and histone H4 are deposited as mono or di-acetylated species. In several subsequent studies, distinct histone H4-acetyl or H3-acetyl species were separated, excised and eluted from an acid urea gel before being micro-sequenced to demonstrate that lysines 4 and 11 of histone H4 (H4K4/H4K11, analogous to H4K5/H4K12 in yeast/humans) [79], and H3K9 and H3K14 [80] are acetylated in *Tetrahymena* associated with new histone synthesis and deposition. A similar approach demonstrated that H4K7, analogous to H4K8 in yeast/humans, is associated with transcription in the MAC [79]. These *Tetrahymena* proteomic studies represent some of the first evidence that histone acetylation occurred in a dynamic and non-random manner. A similar approach was used to demonstrate that MAC H1 (Hho1) possessed five distinct phosphorylation sites in growing but not starved *Tetrahymena* [81]. Identification of the sites permitted their functional analysis through mutation revealing a role for H1 phosphorylation in the regulation of transcription [82,83]. A proteomic study provided early important information concerning histone methylation in *Tetrahymena* and its link to transcription. Histone H3 from ^3^H-AdoMet-labeled MAC was micro-sequenced to demonstrate that lysine 4 is the major site of methylation in MAC in growing cells [84], which was the first correlation between H3K4me and active transcription. 

The development of MS was a powerful tool leading to the identification of new histone PTMs. One study examined PTM profiles on histone H3 from several model organisms, including *Tetrahymena*, yeast, mouse, and human [85]. Whole cell (MAC and MIC) histone H3 was purified by reversed phase chromatography and analyzed by MS. The study found that *Tetrahymena* and budding yeast both possessed a preponderance of H3K4me1, -me2, and -me3 when compared to human cells [85]. *Tetrahymena* was also found to possess H3K56ac [85], a modification that was first reported in *S. cerevisiae* in a MS analysis of its histone PTMs [86,87]. In contrast to budding yeast, *Tetrahymena* was found to have very little H3K79me but was found to contain H3K27me [85]. H3K27me3 was subsequently found to be required for programmed DNA deletion that occurs during MAC development in *Tetrahymena* conjugation [88]. 

Individual and distinct histone PTMs have well established roles in certain biological processes, notably regulation of transcription [89]. A MS approach relying on electron capture dissociation (ECD), a high energy fragmentation technique, was used in combination with a second method termed proton transfer charge reduction (PTR) to identify combinations of histone PTMs on the same MAC H3 molecule associated with transcription [90]. More specifically, the first 50 amino acids of H3 (H3_1–50_) were sequenced MS using ETD/PTR to reveal that H3K27me2 is more correlated with hypo-acetylated H3 and can co-occur with H3K4me1, whereas H3K4me3 was found to be correlated with hyper-acetylated H3 [90]. A related study aimed to correlate histone PTM patterns across 15 different conditions from different physiological states and to mutations of key histone modifying enzymes [91]. Five specific chromatin states were identified; each associated with a set of specific histone PTMs that likely map to distinct functions. To identify MIC specific histone PTMs, MIC were isolated and total histone extracted and separated by reverse-phase HPLC [92]. H3_S_ and H3_F_ were analyzed using pH gradient hydrophilic interaction liquid chromatography (HILIC) directly coupled to high-resolution MS using electron transfer dissociation (ETD) [92]. Both displayed hypo-acetylation, enrichment for H3K27me without H3K4me. Known PTMs found included H3K27me3, and the mitotic PTM histone H3 serine 10 phosphorylation (H3S10ph), were only detected on H3_F_. Additionally, methylation of H3K23, a novel PTM was identified. H3K23me3 in particular was found to be exclusive to H3_F_ and occurred in combination with H3K27me. Functional analysis of the PTM revealed Ezl3 to be the histone methyltransferase (HMT) in addition to having a functional role in meiosis [92]. 

## 8. Chromatin Assembly

AP-MS of histone proteins has allowed for the elucidation of the mechanisms governing chromatin assembly in yeast and human models [93,94,95,96]. Eukaryotic chromatin assembly is a tightly regulated process [97] and new histone chaperones and chromatin modifying enzymes are likely to be identified in future. Owing to the unique biology of nuclear dimorphism involving physical separation of chromatin states, proteomic analysis of chromatin assembly in *Tetrahymena* offers the possibility of gaining novel mechanistic insights as well as potentially identifying new factors with functions in chromatin assembly. 

Proteomic methods have yielded insight into chromatin related processes in *Tetrahymena*. Asf1 is a key generalized H3-H4 chaperone upstream of replication dependent (RD) and independent (RI) chromatin assembly. Much has been learned about Asf1 function by the identification and characterization of Asf1 protein–protein interaction partners [98]. To address the role of *Tetrahymena* Asf1 in chromatin assembly, we engineered a cell line expressing Asf1-FZZ from its endogenous promoter [99]. Two steps of AP were performed both with native elution (TEV protease from IgG and 3xflag peptide from M2-agarose) and analyzed using gel-free LC–MS/MS. To provide statistical rigor to the AP–MS analysis which yielded hundreds of identified proteins, raw MS data were filtered using SAINT (Significance Analysis of INTeractome). SAINT is an algorithm that uses quantitative spectral counts to assign a confidence value to individual protein–protein interactions [100,101]. The algorithm assigns a probability value to an interaction based on data from both control and experimental AP–MS taking into consideration data from biological replicates permitting rigorous discrimination between true and false interactions. Replicate Asf1-FZZ AP-MS were filtered against multiple mock AP-MS from extracts made from untagged cells using SAINT to generate a list of high confidence interacting proteins that included importin β6 (Imbβ6), Nrp1 and three previously uncharacterized Asf1-interacting proteins (AIP1-3). Nrp1 is similar to the *Xenopus* N1/N2 protein, the founding member of a histone chaperone family that also includes human NASP, and budding yeast Hif1. Aip1 and Aip2 do not have similarity to proteins from other organisms and do not contain recognizable domains other than a single coiled coil. A strain was engineered to express Impβ6-FZZ from its chromosomal locus. Indirect IF showed identical localization for Asf1-FZZ and Impβ6-FZZ, namely to both the MAC and MIC but with a more intense signal in the MIC [99]. Reciprocal purification was performed on Impβ6-FZZ suggesting physical interaction with Asf1, Aip1 and Aip2. The Asf1-Impβ6 interaction had been previously predicted from a gene expression analysis examining global gene expression through the *Tetrahymena* life cycle including vegetative growth, various time points during starvation and conjugation [102]. Genes encoding proteins known to interact or to function in complexes show similar expression patterns, indicating that co-ordinate expression with putative genes of known function can identify genes with related functions. Impβ6 was found to be co-expressed with Asf1 with high confidence; the co-expression, physical interaction and similar localization suggests that the two proteins are functionally linked throughout the *Tetrahymena* life cycle [31,102]. Further gene expression network analysis showed clustering of Asf1, and Impβ6 with genes encoding putative HIRA and CAF-1 subunits suggesting that Asf1 most likely functions upstream of RD and RI chromatin assembly in the MAC and MIC [99]. Since Asf1 was not found to co-purify with HIRA or CAF-1 subunits under the conditions used, further work will be needed to determine the mechanism of Asf1 function in *Tetrahymena* chromatin assembly. In human cells, codanin binds Asf1 and negatively regulates importin-4 mediated transport of Asf1–H3–H4 into the nucleus [103]. Although no significant sequence similarity exists between Aip1, Aip2 and human codanin, it is tempting to speculate that the role of Aip1 and Aip2 is related to codanin function, with regulation of differential MAC/MIC import of Asf1-H3-H4. 

We recently reported a proteomic analysis of core histones H2A, H2B and H2A variant H2A.Z (Hv1) [8]. SAINTexpress curated AP-MS data from endogenously tagged H2A-FZZ, H2B-FZZ and Hv1-FZZ strains identified 138 total H2A (Hv1)-H2B co-purifying proteins [8] including members of several highly conserved histone chaperones and chromatin remodeling complexes that have previously been implicated in H2A/H2B and/or variant Hv1 metabolism in humans and yeast, including orthologs of yeast Spt16 and Pob3 (Supt16h and Ssrp1 in humans), the two primary subunits of FACT (facilitates chromatin transcription) complex. FACT was initially named for its activity of reversing a nucleosome-mediated block to RNAPII progression in vitro [104,105]. *Tetrahymena* Spt16 was previously identified using a biochemical approach as associated with the MAC and also microtubules [106]. Our reciprocal AP-MS of Spt16-FZZ identified the ortholog of yeast Pob3 indicating a conserved composition of FACT [8]. Consistent with a role in transcription, several subunits of RNA polymerase also co-purified with Spt16-FZZ. Indirect IF analysis indicated that Spt16-FZZ localizes to both the MIC as well as to the MAC suggesting transcription-independent function(s) for *Tetrahymena* FACT. A novel *Tetrahymena*-specific protein also co-purified with Spt16-FZZ which was named Fimp1 (FACT-interacting mysterious protein 1) whose function remains unexplored. Newly synthesized histone H3 and H4 in human cells carry poly (ADP-ribosylation), a PTM which is poorly understood and has been proposed to help in the proper folding of histones [107]. Early studies reported that *Tetrahymena* histones H2A and H2B are highly ADP-ribosylated [108]. The functional significance and the identity of the Poly (ADP-ribose) polymerase (PARP) have remained unknown. The *Tetrahymena* genome encodes at least 11 putative PARPs and their expression patterns appear to be temporally regulated throughout the life cycle [8]. We identified several PARPs (PARP 1,2,4,5,6) to co-purify with histones H2A (Hv1)-H2B suggesting a potential role for poly(ADP-ribosylation) in *Tetrahymena* histone metabolism [8]. Further studies are required to examine the importance of PARPs to *Tetrahymena* chromatin assembly.

In growing cells, core H2A/H2B localize to MAC and MIC [109,110] whereas Hv1 is only found in MAC [111]. Our AP-MS analysis identified Impβ6 as a high confidence interaction partner of histones H2A-H2B [8]. This result, combined with the previously identified role of Impβ6 in Asf1-based H3/H4 transport [99], suggests that Impβ6 might be a generalized karyopherin for core histone transport pathways. In this respect, it is interesting to note that a different karyopherin, Impβ3, co-purified in Hv1-FZZ AP-MS [8], although it did not pass our stringent statistical threshold. Consistent with a functional link between Hv1 and Impβ3, IF analysis of endogenously tagged Impβ3-FZZ indicated that, like Hv1, Impβ3 exclusively localized to MAC during vegetative growth [112]. Future studies will focus on functional analysis of the identified karyopherins to gain mechanistic insights into *Tetrahymena* chromatin assembly pathways.

## 9. Chromatin Remodeling

Chromatin remodeling (CR) is typically performed by large multi-protein complexes and is required for essential DNA transactions such as transcription, replication, and repair to occur. Mechanisms of CR involve the ATP-dependent histone sliding (e.g., SWI/SNF, and INO80) [113] or selective insertion of histone variants (e.g., SWR complex [94] inserts H2A.Z) in addition to histone PTMs that include acetylation, methylation, and phosphorylation [114,115,116]. Acetylation occurs on lysine residues and is catalyzed by KATs [117] and removed, or ‘erased’, by lysine deacetylases (KDACs) [118]. KATs and KDACs that acetylate/deacetylate histone substrates were initially termed histone acetyltransferases/deacetylase (HATs/HDACs) since histones were their first substrates identified [119]. Histone acetylation occurs either at the nucleosomal level (SAGA [120] and NuA4 [121] complexes) or on histones prior to their deposition into chromatin (Hat1 [122], Rtt109 [123]). Bromodomains (BRD) [124] recognize, or ‘read’ lysine acetylation in order to condense chromatin, regulate transcription or organize protein complexes. Additional protein domains that function in transcription complexes by recognizing some of the diverse histone PTMs include the methyl lysine-recognizing PHD and chromodomains [125,126]. ATP-dependent CR complexes that function in transcription include the SWR complex that exchanges core H2A in the nucleosome for the transcription-friendly histone H2A variant Htz1 [127,128] and the INO80 complex, one function of which is to catalyze the reverse reaction [129].

Proteomic methods using *Tetrahymena* have made significant contributions to understanding fundamental CR mechanisms. The first molecular cloning of the gene encoding a KAT took advantage of *Tetrahymena*’s polyploid MAC specialized for transcription, a source of abundant KAT activity [130]. An acetyltransferase activity in-gel assay was developed [131] to directly identify a catalytically active HAT polypeptide of 55kD from MAC extracts by virtue of its ability to transfer [^3^H]-acetate from [^3^H]-acetyl CoA to core histone H3 incorporated within a polyacrylamide gel [18]. The protein was purified via conventional chromatography using the an in-gel HAT assay and proteolytic analysis resulted in the identification of several internal peptide sequences. Molecular cloning of the gene and comparison of its derived amino acid sequence with available databases revealed it to be highly similar to the yeast protein Gcn5, a transcriptional co-activator implicated in mediating transcriptional activation by acidic activators such as yeast Gcn4p [132,133]. In budding yeast, Gcn5 was found to be the catalyst of the SAGA histone acetyl transferase complex, co-activating transcription by acetylating specific lysine residues in the N-terminus of histone H3 within the nucleosome, which can then serve as a platform to recruit the SWI/SNF complex via the BRD present in SNF2/Brg1 [134]. When recruited to a genomic region, the SWI/SNF complex co-activates transcription by remodeling nucleosomes to make promoter sequences available to be bound by general transcription factors (GTFs) such as TFIID. 

Proteomic methods are being used to begin to elucidate how chromatin remodeling functions in *Tetrahymena*. A SNF2-related gene Brg1^Tt^ was cloned and despite high primary sequence similarity of the protein to that of budding yeast Snf2 and human Brg1 through most of the protein including the ATPase domain, it was not found to possess a BRD [135] initially raising the possibility that SWI/SNF in *Tetrahymena* functions independently of histone acetylation. In order to better understand function of SWI/SNF, AP-MS was performed on a conserved component of SWI/SNF in yeast and human cells, Snf5^Tt^ [7]. MS data resulting from AP of Snf5-FZZ expressed from its native locus was curated using SAINTexpress [136]. The data revealed Snf5 to co-purify with Brg1^Tt^, and *Tetrahymena* orthologs of fundamental SWI/SNF proteins Swi1^Tt^, Swi3^Tt^, and Snf12^Tt^ in addition to four other proteins without clear orthologs in other described SWI/SNF complexes, along with two novel proteins with clear chromatin-interacting domains Saf5 and Ibd1 [7]. Saf5 possesses two tandem plant homeodomains (PHD domain), one function of which is to mediate specific interactions with methylated lysine on histone proteins [137]. Ibd1 possesses a single C-terminal BRD suggesting that the *Tetrahymena* SWI/SNF complex functions downstream of histone acetylation (at least in growing cells) despite the lack of a C-terminal BRD on Brg1/Snf2. Ibd1 was expressed as a recombinant protein with an N-terminal 6x-HIS tag and shown to recognize specific histone acetyl marks associated with the transcriptionally active MAC. Indirect immunofluorescence of Ibd1-FZZ indicated localization to the MAC throughout the *Tetrahymena* life cycle consistent with function in transcription. ChIP-Seq of Ibd1-FZZ demonstrated its enrichment at highly expressed genes. AP-MS of Ibd1-FZZ co-purified not only SWI/SNF members, but components of three additional putative CR complexes, SWR (including orthologs of Swc4, Swr1, Arp6, Swc2, Swc5), SAGA (including orthologs of Ada2 and Gcn5), and an Mll1-related HMT [137]. AP-MS of Hv1-FZZ also co-purified several orthologs of SWR-C subunits consistent with *Tetrahymena* SWR function in Hv1 deposition [8]. Hv1-FZZ AP-MS also co-purified subunits of a possible *Tetrahymena* INO80 complex suggesting that Hv1 localization is dynamically regulated by SWR and INO80 complexes, similar to the case in yeast [138]. Separate reciprocal AP-MS of both Swc4-FZZ and Ada2-FZZ expressed from their respective chromosomal loci revealed that Ibd1 is a distinct component of these complexes and the four complexes do not interact with each other. The Ibd1-SWI/SNF interaction was greatly diminished when AP-MS was performed on Ibd1-FZZ from extracts made from conjugating cells [137]. A model for Ibd1 function was proposed suggesting that Ibd1 may be particularly important to maintain high rates of transcription on highly expressed genes by recognizing one or more specific histone Kac marks that are associated with transcription. Ibd1 could thus act as a scaffold allowing for the recruitment of multiple chromatin-related activities to the region to either further acetylate nearby chromatin (SAGA^Tt^), to remodel nucleosomes (SWI/SNF^Tt^), to deposit transcription-associated histone H2A variant Hv1 (SWR^Tt^), and to di- or tri-methylate histone H3K4 (Atrx3/Set1-like histone methyl transferase [7]. There are 14 additional BRD-containing proteins encoded in the *Tetrahymena* genome [7]. Phylogenetic analysis of *Tetrahymena* BRDs showed that Ibd1 is a member of a five-protein family. AP-MS of the additional family members to determine if regulation of transcription related CR activities is a general feature of this BRD family is in progress. 

Chromodomain helicase DNA-binding proteins (Chd) are a family of SNF2 ATPase chromatin remodelers which mediate key developmentally-regulated processes in eukaryotes [139], initially described in *Tetrahymena* by Fillingham et al. [135]. These proteins are candidate DNA and RNA binding proteins [140]. In vivo and in vitro functional analyses on Chd3 and Chd7 identified diverse macromolecular interactions involved in chromatin dynamics throughout the lifecycle [141]. AP-MS on vegetative and conjugating cell extracts demonstrated that Chd3 interacts with a novel MIZ/SP-RING protein named Miz1, which is a candidate E3 SUMO ligase. Miz1 is likely multiply sumoylated in *Tetrahymena* [141]. We are investigating whether this complex may regulate transcription via epigenetic SUMOylation of chromatin throughout the life cycle.

## 10. Transcription

RNAPII-dependent transcription in eukaryotic cells begins when a transcription factor (TF) binds a specific enhancer DNA and, in conjunction with coactivators, specifies the recruitment of the GTFs and ultimately RNAPII to the promoter of the gene. The composition of RNAPII, GTFs, co-activators and elongation and termination complexes of transcription in *Tetrahymena* remain mostly uncharacterized. *Tetrahymena* also features large-scale transcription of ncRNAs by RNAPII during meiosis that ultimately specify programmed DNA deletion in MAC development [142]. A global MIC-specific nuclear run-on analysis showed that meiotic MIC-specific transcription is biased towards IES DNA, implying that initiation/start-site selection of the MIC-specific ncRNA transcription is regulated and not simply a result of global or random transcription [143,144,145]. The underlying molecular mechanisms regulating the RNAPII-based ncRNA transcription of MIC-specific DNA during meiosis are also mostly unknown. A functional proteomic approach could yield information on the mechanism of transcription of both mRNA during growth and ncRNAs during meiosis. The Mediator transcriptional co-activator complex is widely required for transcription in yeast and human cells. Through AP-MS of conserved Mediator subunit Med31-FZZ [146], we recently identified a divergent *Tetrahymena* Mediator complex. Indirect immunofluorescence of Med31-FZZ demonstrates it to localize to the MAC during growth and development consistent with function as a transcriptional regulator and also to the meiotic MIC implicating it in expression of ncRNAs [146]. 

## 11. Comparative Proteomics

Comparative proteomics has the potential to serve as a powerful tool to study proteome evolution [12,147]. Previous studies have suggested important role(s) for histones [148], histone chaperones [149] and other chromatin-related proteins [150] during eukaryotic evolution. However, these studies focused mainly on Opisthokonts and are limited in their scope due to a lack of experimental data in more divergent model organisms. Ciliates and humans diverged ~1781 million years ago [151] and therefore comparative proteomic analysis of chromatin-related proteins in *Tetrahymena* will be particularly useful to study their role in eukaryotic evolution. Recent small-scale comparative proteomic studies on chromatin-related proteins have provided some interesting observations. 

In humans and in budding yeast, Asf1 has been found to interact with NASP and Importin4 to regulate H3/H4 transport from the cytoplasm to the nucleus [93]. The finding that Asf1 interactions with NASP and Importinβs are conserved in *Tetrahymena* suggest that the regulatory roles of these proteins in H3/H4 transport were acquired early during eukaryotic evolution [99]. This hypothesis motivated further studies and NASP-family proteins were reported to be present throughout the eukaryotes with conserved domain architecture maintained via unusually strong purifying selection [152]. Importantly, the acidic residues shown to be functionally and structurally important for human NASP [153] and budding yeast Hif1 [154] orthologs were found to be highly conserved in *Tetrahymena* Nrp1. It will be interesting to study Nrp1 PPIs during *Tetrahymena* growth and development to gain further insights into its functions. In a comparative proteomic analysis of *Tetrahymena* H2A/H2B and variant Hv1 [8], we identified a set of interacting proteins shared among the three analyzed histones that includes the FACT complex, as well as H2A- or Hv1-specific chaperones. The co-purification of an Npm1 ortholog with H2A/H2B highlights the utility of *Tetrahymena* as an excellent microbial model organism to study chromatin regulatory layers. Npm1 is an H2A/H2B histone chaperone which has been shown to function in a myriad of chromatin-, transcription- and cell cycle-regulatory pathways [155]. Owing to its loss in widely used microbial model organisms such as budding yeast, most previous studies have been restricted to cultured cells. Our comparative analysis indicated that the *Tetrahymena* ortholog is structurally conserved and appears to be more closely related to human Npm1 instead of human Npm2-3 proteins, and hence was named as conserved Npm1-like 1 (cNpl1). Although detailed analysis awaits further studies, initial observations suggested that cNpl1 exclusively localized to MAC during vegetative growth, and may have a role in cell cycle regulation during early developmental stages in *Tetrahymena*. We suggest that histones are subject to an ancient network of chaperones which were likely present in the last common ancestor of eukaryotes [8]. Comparative evolutionary proteomic analysis of core histones H3/H4 and variant H3.3 (Hv2) in addition to MAC and MIC linker histones will be instrumental to provide a comprehensive functional annotation of *Tetrahymena* chromatin assembly pathways and potentially reveal very useful information regarding eukaryotic evolution. 

In addition to the above studies, some large-scale comparative proteomic analyses have been reported in *Tetrahymena*. An MS-based high-throughput phosphoproteomic analysis was carried out to identify phosphorylation sites and phosphoproteins in *Tetrahymena* during growth, starvation and three different developmental stages [156]. Samples from different stages of the *Tetrahymena* life cycle were mixed and a comparative phosphoproteomics analysis through developmental stages was not possible. However, a comparative analysis of *Tetrahymena*, *Plasmodium falciparum* and human phosphoproteomes was carried out and the residue composition surrounding the phosphorylation sites was determined. It was observed that *Tetrahymena* shared some conserved features with *P. falciparum* and humans in the properties of amino acid residues surrounding the phosphorylation sites. For example, P at position +1 and R at position −3 for the serine phosphorylation site were consistently overrepresented in all three species. Furthermore, consistent with previous studies [157], the authors found that the most enriched residues in *Tetrahymena* phosphopeptides were significantly overrepresented in intrinsically disordered regions of the proteins. 

## 12. New Proteomic Technologies for the Characterization of *Tetrahymena* Nuclear Proteins

Innovation drives progress in proteomics and a variety of new technologies and methods are poised to allow new discoveries of *Tetrahymena* chromatin structure and function. For instance, proximity biotinylation (BioID), first reported in 2012 [158], was found to be particularly effective in mapping membraneless organelles and proteins of poor solubility [159,160,161,162]. It is a complementary approach to AP-MS to map PPIs of nuclear proteins including chromatin associated proteins [163]. BioID was successfully used in *Tetrahymena* for the study of cilia proteins [164,165] and is a promising tool to be implemented to further the study of nuclear proteins including chromatin associated proteins. We are employing this approach to address questions about the membraneless organelle known as the Conjusome present at a specific time during conjugation and important in developmentally programmed genome reorganization and irreversible genome silencing [166]. Two proteins, Pdd1 and Tcd1 have been found associated with the *Tetrahymena* Conjusome [167]. Attempts to identify proteins interacting with Pdd1 using AP-MS at a time in conjugation when the Conjusome is present have not been successful (Pearlman et al., unpublished observations). BioID is ideal for analysis of interactomes of Pdd1 and Tcd1. This will provide important information about genome reorganization and irreversible genome silencing in *Tetrahymena* and will also provide insights into the now very important area of phase separations and membraneless organelles identified and under vigorous study in many systems such as P-Bodies [168], stress granules [168], and Cajal bodies [169]. 

Other proteomics approaches that have the potential to enhance our understanding of *Tetrahymena* chromatin structure and functions, are methods allowing for the identification of proteins associated with a specific genomic locus. The development of effective genome targeting modules (e.g., Cas9) has allowed numerous groups to deploy these types of approaches. For instance, Myers et al. and Gao et al. employed an inactivated Cas9 (dCas9) enzyme fused to an engineered ascorbate peroxidase (APEX2) to biotinylate proteins surrounding a genomic locus of interest [170,171]. Subsequently, the biotinylated proteins could be purified using streptavidin resin and quantified using MS. One could expect these approaches to significantly contribute to the characterization of the proteins involved in regulation of nuclear function in *Tetrahymena* during its life cycle including during its programmed DNA rearrangements.

## 13. Conclusions

Proteomic analysis has advanced knowledge of nuclear structure/function in many systems. Studies in *Tetrahymena* have been detailed and comprehensive and, due mainly to the unique nuclear dimorphism of ciliates, have in many cases informed studies in other eukaryotes. Proteomic studies will continue to yield new information concerning nuclear functions in *Tetrahymena* particularly in several relatively unexplored areas such as global transcription of ncRNAs.

## Figures and Tables

**Figure 1 genes-10-00333-f001:**
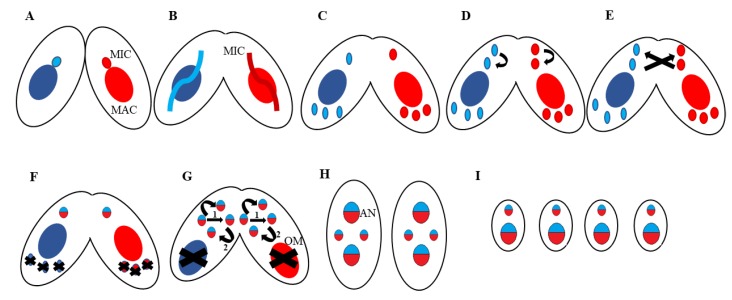
Conjugation (Mating) and Nuclear Development in *T.thermophila*. (**A**) Cell adhesion occurs at 1.5 h. (**B**) Meiosis (Prophase I) occurs at 3 h where the micronucleus (MIC) crescent is formed. (**C**) At 4.5 h, four gametic nuclei are formed. (**D**) Duplication of one of the posterior gametic nuclei occurs. (**E**) At 6 h, nuclear exchange followed by fusion of the gametic nuclei occurs, followed by, (**F**) Formation of the zygotic nucleus and elimination of the other three gametic nuclei. (**G**) At 8 h two rounds of zygotic duplication occurs, old macronucleus (MAC) (OM) degradation and new developing MAC (Anlagen (AN)) begins. (**H**) 2 Anlagen 2 MIC stage. (**I**) At 10 h separation of exconjugants occurs and at this point is when chromosome fragmentation, Internal Eliminated Sequences (IES) deletion and DNA amplification starts (~45C). At the end of conjugation there are four daughter cells.

**Figure 2 genes-10-00333-f002:**
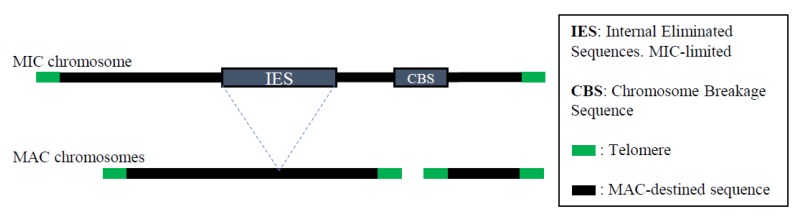
Extensive Programmed DNA Rearrangements in *Tetrahymena*. These events occur during MAC development where ~10–15% of MIC genome is deleted. See text for additional details.

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
