# Peer review of "Functional Proteomics of Nuclear Proteins in Tetrahymena thermophila: A Review"

_genes, 2019, doi:10.3390/genes10050333_

Reviewer 1 Report

Tetrahymena has proven to be an important model system in the field of chromatin research, and has more to contribute in other areas. The application of advanced proteomic tools will be important for future discoveries. The authors have presented a thorough, and accurate review of the proteomics projects that have been completed, are ongoing, or could contribute to other fields in the future. This review is unique, in that, as far as I know, this information on nuclear proteins has not been consolidated since before the Tetrahymena genome was sequenced. The experiments explained and proposed will inform other researchers working on ciliates about the approaches available to address protein interactions. This manuscript should also be of use to researchers studying other eukaryotic model systems, in designing protein purification methods that pinpoint the question being addressed. I have a few small writing suggestions that may help the non-ciliate reader. 

line 70 add "germline" before micronucleus 

line 77 what "irreversible gene silencing" in the MAC is this referring to? 

line 89 Figure 1 legend add "MIC" before crescent

line 92 add "posterior" before gametic

line 194 wording of "and new developing MAC (Anlagen)" use this phrase in the legend for Figure 1, for clarity as early as possible.

line 211 rewrite this sentence stating more clearly that the 4 samples - two vegetative, and two starved, were compared to the anlagen sample 

line 215  "three MAC preps" - specify anlagen separate from MAC

line 223 Figure 2 key - add black line labelled MAC-destined sequence

line 267 move "PAZ/PIWI domain (PPD)" to line 225

line 308 begin new paragraph after [84]. 

line 360 a better reference or additional reference is Xiong J, Lu Y, Feng J, et al. Tetrahymena functional genomics database (TetraFGD): an integrated resource for Tetrahymena functional genomics. Database (Oxford). 2013;2013:bat008. Published 2013 Mar 12. doi:10.1093/database/bat008

Author Response

line 70 add "germline" before micronucleus

·         Done

line 77 what "irreversible gene silencing" in the MAC is this referring to?

·         Corrected and clarified.

line 89 Figure 1 legend add "MIC" before crescent

·         Done

line 92 add "posterior" before gametic

·         Done

line 194 wording of "and new developing MAC (Anlagen)" use this phrase in the legend for Figure 1, for clarity as early as possible.

·         Done

line 211 rewrite this sentence stating more clearly that the 4 samples - two vegetative, and two starved, were compared to the anlagen sample

·         Done. Deleted ‘four’ to clarify that samples included MAC and MIC from vegetative and starved cells and Anlagen.

·         line 215  "three MAC preps" - specify anlagen separate from MAC

Sentence modified accordingly     O.K. and I have made a few other additional very minor grammatical changes to the sentence.

line 223 Figure 2 key - add black line labelled MAC-destined sequence

·         done

line 267 move "PAZ/PIWI domain (PPD)" to line 225

·         done

line 308 begin new paragraph after [84].

·         done

line 360 a better reference or additional reference is Xiong J, Lu Y, Feng J, et al. Tetrahymena functional genomics database (TetraFGD): an integrated resource for Tetrahymena functional genomics. Database (Oxford). 2013;2013:bat008. Published 2013 Mar 12. doi:10.1093/database/bat008

·         added

Reviewer 2 Report

This review of Functional Proteomics of Nuclear Proteins in Tetrahymena thermophila is very thoroughly researched and well organized. The sections will provide readers with a comprehensive review on specific topics of interest.

- One concern about the manuscript is at line 207, where the authors discuss results from one of their own experiments which is current unpublished. Typically new results are not included in a review paper, and if this is the first time that these results are being presented, then at least a table of the results should be presented so that readers have some level of access to the results besides what's being described here. If these results will be presented in a future separate publication, then they should not be included here.

- Some mention of how the wide array of proteomics datasets are complemented by genomic and transcriptomic resources in the Tetrahymena thermophila community would help to frame where this review fits into the bigger picture. For example, there is the tetraFGD database (PMID:23482072) and some transcriptomic datasets (eg PMID:22347391), but because of its  unique biology, proteomic studies offer much more insight into this particular organism.

Author Response

One concern about the manuscript is at line 207, where the authors discuss results from one of their own experiments which is current unpublished. Typically new results are not included in a review paper, and if this is the first time that these results are being presented, then at least a table of the results should be presented so that readers have some level of access to the results besides what's being described here. If these results will be presented in a future separate publication, then they should not be included here.

·         As this is a review, we do not wish to present new data and do not wish to include a table of results. Although ‘unpublished’ these data have been presented and discussed at a number of international conferences. We have presented the text in this section to clearly indicate this as an approach that will be of interest to the community and we are confident that this is appropriate for a review of this type and will add value to this review and its use to the ciliate and broader communities.

Some mention of how the wide array of proteomics datasets are complemented by genomic and transcriptomic resources in the Tetrahymena thermophila community would help to frame where this review fits into the bigger picture. For example, there is the tetraFGD database (PMID:23482072) and some transcriptomic datasets (eg PMID:22347391), but because of its  unique biology, proteomic studies offer much more insight into this particular organism.

·         Added some text to the bottom of P2 to incorporate this suggestion.